# Distributional Robustness with IPMs and links to Regularization and GANs

**Hisham Husain**
The Australian National University & Data61
`hisham.husain@anu.edu.au`

## Abstract

Robustness to adversarial attacks is an important concern due to the fragility of deep neural networks to small perturbations and has received an abundance of attention in recent years. Distributional Robust Optimization (DRO), a particularly promising way of addressing this challenge, studies robustness via divergence-based uncertainty sets and has provided valuable insights into robustification strategies such as regularisation. In the context of machine learning, majority of existing results have chosen $f$-divergences, Wasserstein distances and more recently, the Maximum Mean Discrepancy (MMD) to construct uncertainty sets. We extend this line of work for the purposes of understanding robustness via regularization by studying uncertainty sets constructed with Integral Probability Metrics (IPMs) - a large family of divergences including the MMD, Total Variation and Wasserstein distances. Our main result shows that DRO under *any* choice of IPM corresponds to a family of regularization penalties, which recover and improve upon existing results in the setting of MMD and Wasserstein distances. Due to the generality of our result, we show that other choices of IPMs correspond to other commonly used penalties in machine learning. Furthermore, we extend our results to shed light on adversarial generative modelling via $f$-GANs, constituting the first study of distributional robustness for the $f$-GAN objective. Our results unveil the inductive properties of the discriminator set with regards to robustness, allowing us to give positive comments for a number of existing penalty-based GAN methods such as Wasserstein-, MMD- and Sobolev-GANs. In summary, our results intimately link GANs to distributional robustness, extend previous results on DRO and contribute to our understanding of the link between regularization and robustness at large.

## 1 Introduction

Robustness to adversarial attacks is an important concern due to the fragility of deep neural networks to small perturbations and has received an abundance of attention in recent years [21, 50, 31]. Distributionally Robust Optimization (DRO), a particularly promising way of addressing this challenge, studies robustness via divergence-based uncertainty sets and considers robustness against shifts in distributions. To see this more clearly, for some space $\Omega$, model $h : \Omega \to \mathbb{R}$ and training data $\hat{P}$ with empirical loss $\mathbb{E}_{x \sim \hat{P}}[l_f]$, DRO when applied to machine learning studies the objective $\sup_{Q \in \mathcal{U}} \mathbb{E}_{x \sim Q}[l_f]$ where $\mathcal{U} = \left\{ Q : d(Q, \hat{P}) \leq \varepsilon \right\}$ for a given divergence $d$ and $\varepsilon > 0$ that characterize the adversary. Work along this line has shown that this objective is upper bounded by the empirical loss $\mathbb{E}_{x \sim \hat{P}}[l_f]$ plus a penalty term that plays the role of a regularizer, consequently providing formal connections and valuable insights into regularization as a robustification strategy [22, 27, 36, 5, 14, 11].

The choice of $d$ is crucial as it highlights the strength and nature of robustness we desire, and different choices yield differing penalties. It has been shown that minimizing the distributionally robust objective when $d$ is chosen to be an $f$-divergence is roughly equivalent to variance regularization [22, 27, 36]. However, there is a problem with this choice of $d$, as highlighted in [48]: every distribution in the uncertainty set is required to be absolutely continuous with respect to $P$. This is particularly problematic in the case when $P$ is empirical since every distribution in $\mathcal{U}$ will be finitely supported, meaning that the population distribution will not be contained as it is typically continuous.

Choosing the Wasserstein distance as $d$ is a typical antidote for this problem, and much work has been invested in this direction, explicating connections to Lipschitz regularization [20, 10, 44, 42, 11]. More recently, uncertainty sets based on the kernel Maximum Mean Discrepancy (MMD) were investigated to address concerns with the $f$-divergence and discovered links to regularization with Hilbert space norms. Both the Wasserstein distance and MMD are part of a larger family of divergences referred to as Integral Probability Metrics (IPM) [35], which are characterized by a set of functions $\mathcal{F}$, and include other metrics such as the Total Variation distance and the Dudley Metric [47].

In this work, we generalize these results and study DRO for uncertainty sets induced by the Integral Probability Metric (IPM) for *any* set of functions $\mathcal{F}$. We present an identity which links distributional robustness under these uncertainty sets $\mathcal{U}_{\mathcal{F}}$, to regularization under a new penalty $\Lambda_{\mathcal{F}}$. Our identity takes the form

$$\boxed{\sup_{Q \in \mathcal{U}_{\mathcal{F}}} \int_{\Omega} h\, dQ = \int_{\Omega} h\, dP + \Lambda_{\mathcal{F}}(h)} \tag{1}$$

The appeal of this result is that it reduces the infinite-dimensional optimization on the left-hand side into a penalty-based regularization problem on the right-hand side. We study properties of this penalty and show that it can be upper bounded by another term, $\Theta_{\mathcal{F}}$, which recovers and improves upon existing penalties when $\mathcal{F}$ is chosen to coincide with the MMD and Wasserstein distances. Our result, however, holds in much more generality, allowing us to derive new penalties by considering other IPMs such as the Total Variation, Fisher IPM [33], and Sobelov IPM [32]. We find that these new penalties are related to existing penalties in regularized critic losses [51] and manifold regularization [4], permitting us to provide untried robustness perspectives for existing regularization schemes. Furthermore, most work in this direction takes the form of upper bounds, and although working with $\Theta_{\mathcal{F}}$ reduces (1) into an inequality, we present a necessary and sufficient condition such that $\Lambda_{\mathcal{F}}$ coincides with $\Theta_{\mathcal{F}}$, yielding equality. This condition reveals an intimate connection between distributional robustness and regularized binary classification.

We then apply our result to understanding the distributional robustness of Generative Adversarial Networks (GANs), a popular method for modelling distributions that learn a model $Q$ by utilizing a set of discriminators $D$ that try to distinguish $Q$ from $P$ (the training data). This is particularly relevant for the robustness community since lines of work [53, 9, 58, 57, 28, 26, 39, 45, 46, 24, 55, 40] implement GANs as a robustifying mechanism by training a binary classifier on the learned GAN distribution. Our analysis applies to the $f$-GAN objective [37] - a loss that subsumes many existing GAN losses. This is, to the best of our knowledge, the first analysis of robustness for $f$-GANs with respect to divergence-based uncertainty sets. The main insight of our result is the advocation of regularized discriminators when training GANs. In particular, we show that the generative distribution learned using regularized discriminators gives guarantees on the worst-case perturbed distribution (robustness). Our findings complement existing empirical benefits of regularized discriminators such as the MMD-GAN [29, 2, 6], Wasserstein-GAN [3, 23], Sobelov-GAN [32], Fisher-GAN [33] and other penalty-based GANs [51].

Our contributions come in three Theorems, where the first two concern DRO with IPMs (Section 3) and the third is an extension to understanding GANs (Section 4):
▷ **(Theorem 1)** An identity for distributional robustness using uncertainty sets induced by any IPM. Our result tells us that this is *exactly* equal to regularization with a penalty $\Lambda_{\mathcal{F}}$. We show that this penalty can be upper bounded by another penalty $\Theta_{\mathcal{F}}$ which recovers existing work when the IPM is set to the MMD and Wasserstein distance, tightening these results. Since our result holds in much more generality, we derive penalties for other IPMs such as the Total Variation, Fisher IPM, and Sobelov IPM, and draw connections to existing methods.
▷ **(Theorem 2)** A necessary and sufficient condition under which the penalties $\Lambda_{\mathcal{F}}$ and $\Theta_{\mathcal{F}}$ coincide. It turns out this condition is linked to regularized binary classification and is related to critic losses

appearing in penalty-based GANs. This allows us to give positive results for work in this direction, along with drawing a link between regularized binary classification and distributional robustness.

▷ **(Theorem 3)** A result that characterizes the distributional robustness of the $f$-GAN objective showing that the discriminator set plays an important part for the robustness of a GAN. This is, to the best of our knowledge, the first result on divergence-based distributional robustness of $f$-GANs. Our result allows us to provide a novel perspective for several existing penalty-based GAN methods such as Wasserstein-, MMD-, and Sobelov-GANs.

## 2 Preliminaries

### 2.1 Notation

We will use $\Omega$ to denote a compact Polish space and denote $\Sigma$ as the standard Borel $\sigma$-algebra on $\Omega$ and $\mathbb{R}$ will denote the real numbers. We use $\mathscr{F}(\Omega, \mathbb{R})$ to denote the set of all bounded and measurable functions mapping from $\Omega$ into $\mathbb{R}$ with respect to $\Sigma$, $\mathscr{B}(\Omega)$ to be the set of finite signed measures and the set $\mathscr{P}(\Omega) \subset \mathscr{B}(\Omega)$ will denote the set of probability measures. For any additive monoid $X$, a function $f : X \to \mathbb{R}$ is subadditive if $f(x + x') \leq f(x) + f(x')$ and the *infimal convolution* between two functions $f : X \to \mathbb{R}$ and $g : X \to \mathbb{R}$ is another function given by $(f \barwedge g)(x) = \inf_{x' \in X} (f(x') + g(x - x'))$. For any proposition $\mathscr{I}$, the inversion bracket is $[\![\mathscr{I}]\!] = 1$ if $\mathscr{I}$ is true and 0 otherwise. We say a set of functions $\mathcal{F}$ is even if $h \in \mathcal{F}$ implies $-h \in \mathcal{F}$. For a function $h \in \mathscr{F}(\Omega, \mathbb{R})$ and metric $c : \Omega \times \Omega \to \mathbb{R}$, the Lipschitz constant of $h$ (w.r.t $c$) is $\text{Lip}_c(h) = \sup_{\omega, \omega' \in \Omega} |h(\omega) - h(\omega')| / c(\omega, \omega')$ and $\|h\|_\infty := \sup_{\omega \in \Omega} |h(\omega)|$. For any set of functions $\mathcal{F} \subseteq \mathscr{F}(\Omega, \mathbb{R})$, we use $\overline{\text{co}}(\mathcal{F})$ to denote the closed convex hull of $\mathcal{F}$. For a function $h \in \mathscr{F}(\Omega, \mathbb{R})$ and measure $\mu \in \mathscr{P}(\Omega)$, we use $\text{Var}_\mu(h) = \mathbb{E}_\mu[h^2] - \mathbb{E}_\mu[h]^2$ to denote the variance of $h$ under $\mu$.

### 2.2 Background and Related Work

We will focus our discussion around Distributionally Robust Optimization (DRO) [41] and its use for understanding machine learning. For a given reference distribution $P$, which is typically the training data in machine learning, the neighbourhood takes the form $\{Q : d(Q, P) \leq \varepsilon\}$ for some divergence $d$ and $\varepsilon > 0$ that characterize the nature and budget of robustness. In the context of machine learning, the most popular choices of $d$ studied thus far are the $f$-divergences [5, 13, 27], Wasserstein distance [16, 1, 7] and the kernel Maximum Mean Discrepancy (MMD) [48]. For two distributions $P, Q$, the $f$-divergence is $d_f(P, Q) = \int_\Omega f(dP/dQ)dQ$ and the main advancement regarding $f$-divergences, centered around $\chi^2$-divergence, is the connection to variance regularization [22, 27, 36]. This is appealing since it reflects the classical bias-variance trade-off. In contrast, variance regularization also appears in our results, under the choice of $\mu$-Fisher IPM. One of the drawbacks of using $f$-divergences as pointed out in [48], is that the uncertainty set induced by $f$-divergences contains only those distributions that share support (since we require absolute continuity) and thus will typically not include the population distribution. The Wasserstein distance is commonly antidotal for these problems since it is defined between distributions that do not share support and DRO results have been developed for this direction, with the main results showing links to Lipschitz regularization [20, 10, 44, 42, 11]. Another distance used to remedy this problem is the Maximum Mean Discrepancy, which has been studied in [48] and shown connections to Hilbert space norm regularization and kernel ridge regression. Since both of these are Integral Probability Metrics (IPMs) [35], it is natural to study uncertainty sets generated by general IPMs:

**Definition 1 (Integral Probability Metric)** *For any $\mathcal{F} \subseteq \mathscr{F}(\Omega, \mathbb{R})$, the ($\mathcal{F}$-)Integral Probability Metric between $P, Q \in \mathscr{P}(\Omega)$ is*

$$d_\mathcal{F}(P, Q) := \sup_{h \in \mathcal{F}} \left( \int_\Omega h dP - \int_\Omega h dQ \right).$$

The IPM is characterized by a set $\mathcal{F}$ and if $\mathcal{F}$ is even, then $d_\mathcal{F}$ is symmetric. One should note that we have an intersection between IPMs and $f$-divergence when $\mathcal{F} = \{h : \|h\|_\infty \leq 1\}$ and $f(t) = |t - 1|$, which corresponds to the Total Variation. Other cases when they intersect have been thoroughly pursued in [47]. Another interesting case is the 1-Wasserstein distance, which is realized when

| IPM | $\mathcal{F}$ | $\Theta_{\mathcal{F}}(h)$ |
|---|---|---|
| Wasserstein Distance | $\{h : \mathrm{Lip}_c(h) \leq 1\}$ | $\mathrm{Lip}_c(h)$ |
| Maximum Mean Discrepancy | $\{h : \|h\|_k \leq 1\}$ | $\|h\|_k$ |
| Total Variation | $\{h : \|h\|_\infty \leq 1\}$ | $\|h\|_\infty$ |
| Dudley Metric | $\{h : \|h\|_\infty + \mathrm{Lip}_c(h) \leq 1\}$ | $\|h\|_\infty + \mathrm{Lip}_c(h)$ |
| $\mu$-Sobelov IPM | $\left\{h : \mathbb{E}_{\mu(X)}\left[\|\nabla h(x)\|^2\right] \leq 1\right\}$ | $\sqrt{\mathbb{E}_{\mu(X)}\left[\|\nabla h(X)\|^2\right]}$ |
| $\mu$-Fisher IPM | $\left\{h : \mathbb{E}_{\mu(X)}\left[h^2(X)\right] \leq 1\right\}$ | $\sqrt{\mathbb{E}_{\mu(X)}\left[h^2(X)\right]}$ |

$\mathcal{F} = \{h : \mathrm{Lip}_c(h) \leq 1\}$ for some ground metric $c : \Omega \times \Omega \to \mathbb{R}$ [52]. Table 1 contains other known choices of IPMs. As the IPM can be viewed as matching moments specified by $\mathcal{F}$, there is similar work which considers uncertainty sets that match the first and second moment such as [12]. In the context of machine learning our work is, to the best of our knowledge, the first study of the general IPM to understand regularization. Outside this realm, there exist pursuits to study structural properties of IPM-based uncertainty sets such as invariance [43]. While these are important to understand, they, however, do not give immediate consequences for machine learning.

## 3 Distributional Robustness

In this section, we first introduce the uncertainty set and two complexity measures that form building blocks of the main penalty term $\Lambda_{\mathcal{F}}$ (as appearing in Equation 1), then proceed to the main distributional robustness Theorem.

**Definition 2** *For any $\mathcal{F} \subseteq \mathscr{F}(\Omega, \mathbb{R})$, $P \in \mathscr{P}(\Omega)$, the $\mathcal{F}$-ball centered at $P$ with radius $\varepsilon$ is defined to be $B_{\varepsilon, \mathcal{F}}(P) = \{Q \in \mathscr{P}(\Omega) : d_{\mathcal{F}}(Q, P) \leq \varepsilon\}$.*

We now introduce a complexity measure that will be of central importance when defining the penalty: For a function set $\mathcal{F} \subseteq \mathscr{F}(\Omega, \mathbb{R})$ and function $h \in \mathscr{F}(\Omega, \mathbb{R})$, we set $\Theta_{\mathcal{F}}(h) := \inf\{\lambda > 0 : h \in \lambda \cdot \overline{\mathrm{co}}(\mathcal{F})\}$. This quantity represents the smallest lambda that multiplicatively stretches the set $\overline{\mathrm{co}}(\mathcal{F})$ until it contains $h$. We illustrate this geometrically in Figure 1 for a non-convex case of $\mathcal{F}$ and present examples of $\Theta_{\mathcal{F}}$ in Table 1.

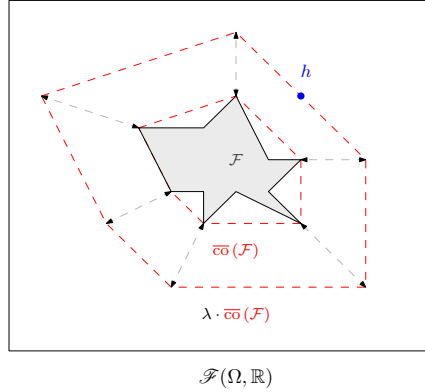

Figure 1: $\Theta_{\mathcal{F}}(h)$ is the smallest multiplicative factor $\lambda$ required to stretch the convex hull of $\mathcal{F}$ until $h$ is contained.

The second complexity measure depends on a distribution $P \in \mathscr{P}(\Omega, \mathbb{R})$ and is defined as $J_P(h) = \sup_{\nu \in \mathscr{P}(\Omega)} \int_\Omega h \, d\nu - \int_\Omega h \, dP$. Note that if $h$ reaches its maximum at some $\omega^* \in \Omega$ then $J_P(h)$ will be smaller if $P$ is concentrated around $\omega^*$. We now present the main penalty, which is infimal convolution of these two complexity measures.

**Definition 3 ($\mathcal{F}$-Penalty)** *For any $\mathcal{F} \subseteq \mathscr{F}(\Omega, \mathbb{R})$, $h \in \mathscr{F}(\Omega, \mathbb{R})$ and $\varepsilon > 0$, the $\mathcal{F}$-penalty $\Lambda_{\mathcal{F}, \varepsilon} : \mathscr{F}(\Omega, \mathbb{R}) \to [0, \infty]$ is*

$$\Lambda_{\mathcal{F}, \varepsilon}(h) = (J_P \,\overline{\star}\, \varepsilon\Theta_{\mathcal{F}})(h),$$

*where $J_P(h) = \sup_{\nu \in \mathscr{P}(\Omega)} \int_\Omega h \, d\nu - \int_\Omega h \, dP$ and $\overline{\star}$ is the infimal convolution operator.*

The infimal convolution is central in convex analysis since it is the analogue of addition in the convex dual space [49]. We now present the main theorem, which links this penalty to distributional robustness via $\mathcal{F}$-uncertainty sets and discuss further the role of this penalty.

**Theorem 1** *Let $\mathcal{F} \subseteq \mathscr{F}(\Omega, \mathbb{R})$ and $P \in \mathscr{P}(\Omega)$. For any $h \in \mathscr{F}(\Omega, \mathbb{R})$ and for all $\varepsilon > 0$*

$$\sup_{Q \in B_{\varepsilon, \mathcal{F}}(P)} \int_\Omega h dQ = \int_\Omega h dP + \Lambda_{\mathcal{F}, \varepsilon}(h).$$

**Proof** (Sketch, full proof in Supplementary material) We can rewrite the constraint over $B_{\varepsilon, \mathcal{F}}(P)$ with the use of a dual variable which leads to a min-max equation. Using generalized minimax theorems [17] and compactness of the set of probability measures, we are able to swap the min-max and solve the inner min using classical results in convex analysis [38], yielding the statement of the theorem. ∎

The result allows us to turn the infinite-dimensional optimization on the left-hand side into a familiar penalty-based regularization objective, and we remark that there is no restriction on the choice of $\mathcal{F}$. To see the effect of $\Lambda_{\mathcal{F}, \varepsilon}$, notice that by definition of $\bar{\star}$ we have

$$\Lambda_{\mathcal{F}, \varepsilon}(h) = \inf_{\substack{h_1, h_2 \\ h_1 + h_2 = h}} \left( J_P(h_1) + \varepsilon \Theta_{\mathcal{F}}(h_2) \right),$$

which means this penalty finds a decomposition of $h$ into $h_1, h_2$ so that the two penalties $J_P(h_1)$ and $\varepsilon \Theta_{\mathcal{F}}(h_2)$ are controlled. Notice that any decomposition gives an upper bound, and this is precisely how we will show links and tighten existing results. We will then present a necessary and sufficient condition under which $\Lambda_{\mathcal{F}, \varepsilon}(h) = \varepsilon \Theta_{\mathcal{F}}(h)$. This condition plays a fundamental role in linking robustness to regularization and unlike majority of existing results, yields an *equality*.

To see the applicability of the result, consider the supervised learning setup: We have an input space $\mathcal{X}$, output space $\mathcal{Y}$, and a loss function $l : \mathcal{Y} \times \mathcal{Y} \to \mathbb{R}$ which measures performance of a hypothesis $g : \mathcal{X} \to \mathcal{Y}$ on a sample $(x, y)$ with $l(g(x), y)$. In this case, we set $\Omega = \mathcal{X} \times \mathcal{Y}$, $P$ to be the available data, and $h = l(g(x), y)$:

$$\sup_{Q \in B_{\varepsilon, \mathcal{F}}(P)} \int_\Omega l(g(x), y) dQ(x, y) = \underbrace{\int_\Omega l(g(x), y) dP(x, y)}_{\text{data fitting term}} + \underbrace{\Lambda_{\mathcal{F}, \varepsilon}(l(g(x), y))}_{\text{robustness penalty}}.$$

The first term is interpreted as a data fitting term, while the second term is a penalty term that ensures robustness of $g$. We remark that upper bounds are still favourable in the application of supervised learning, which we will now discuss.

To generate our first upper bound, consider the following decomposition: $h_1 = b$ and $h_2 = h - b$ for some $b \in \mathbb{R}$, yielding the following Corollary.

**Corollary 1** *Let $\mathcal{F} \subseteq \mathscr{F}(\Omega, \mathbb{R})$ and $P \in \mathscr{P}(\Omega)$. For any $h \in \mathscr{F}(\Omega, \mathbb{R})$ and for all $\varepsilon > 0$*

$$\sup_{Q \in B_{\varepsilon, \mathcal{F}}(P)} \int_\Omega h dQ \leq \int_\Omega h dP + \varepsilon \inf_{b \in \mathbb{R}} \Theta_{\mathcal{F}}(h - b).$$

We will show that Corollary 1 recovers or tightens main results, and holds in much more generality since we may choose *any* set $\mathcal{F}$. The choice of $\mathcal{F}$ is important to our notion of uncertainty as it captures the moments we are interested in, and there is a natural trade-off between picking $\mathcal{F}$ to be too large or too small, which we illustrate with extreme cases. Consider the largest possible set $\mathcal{F} = \mathscr{F}(\Omega, \mathbb{R})$, under which the uncertainty set of distributions, $B_{\varepsilon, \mathcal{F}}(P) = \{P\}$ is a singleton for all $\varepsilon > 0$. This is indeed reflected on the right hand side of Corollary 1, noting that such a strong set $\mathcal{F}$ yields $\Theta_{\mathcal{F}}(h) = 0$ for any $h \in \mathscr{F}(\Omega, \mathbb{R})$. On the other hand, if we pick $\mathcal{F} = \{f(x) = k : k \in \mathbb{R}\}$ to be the set of constants, which is a rather restrictive set, then the uncertainty ball of distributions is the largest it can be $B_{\varepsilon, \mathcal{F}} = \mathscr{P}(\Omega)$ since $d_{\mathcal{F}}(Q, P) = 0$ for all $Q \in \mathscr{P}(\Omega)$. We now focus on non-trivial settings of $\mathcal{F}$, showing that $\Theta_{\mathcal{F}}$ recovers and improves upon familiar existing penalties.

(a) **(Wasserstein Distance)** $\mathcal{F} = \{h : \text{Lip}_c(h) \leq 1\}$. The penalty is $\Theta_{\mathcal{F}}(h) = \text{Lip}_c(h)$, and Corollary 1 recovers the intuition of Lipschitz regularized networks as presented in [20, 10, 44, 42, 8, 11]. However, the penalty in the original theorem $\Lambda_{\mathcal{F}, \varepsilon}$ is tighter. To see this by example, consider $\Omega = \mathbb{R}$, $P$ a normal distribution centered at 0 with variance $\sigma > 0$, $h(t) = \sin 2t + t$ and $\varepsilon = 1$. Note that $\varepsilon \text{Lip}_c(h) = 3$ however $h$ can be decomposed into $h_1 = \sin 2t$ and $h_2 = t$ with $J_P(h_1) = 1$ and $\varepsilon \text{Lip}_c(h_2) = 1$. Hence we have $\Lambda_{\mathcal{F}, \varepsilon}(h) \leq 2 < 3 = \varepsilon \text{Lip}_c(h)$.

(b) **(Maximum Mean Discrepancy)** $\mathcal{F} = \{h : \|h\|_k \leq 1\}$ where $k : \Omega \times \Omega \to \mathbb{R}$ is a positive definite characteristic kernel and $\|\cdot\|_k$ is the Reproducing Kernel Hilbert Space (RKHS) norm induced by $k$ [34]. For $h$ in the RKHS, the penalty can be bounded by $\Lambda_{\mathcal{F},\varepsilon}(h) \leq \inf_{b \in \mathbb{R}} \|h - b\|_k$. This tightens the existing work on MMD DRO [48, Corollary 3.2] when $b = 0$.

(c) **(Total Variation)** $\mathcal{F} = \{h : \|h\|_\infty \leq 1\}$. Our result tells us that the penalty upper bounded with $\Lambda_{\mathcal{F},\varepsilon}(h) \leq \inf_{b \in \mathbb{R}} \|h - b\|_\infty$, which is tighter than taking $\|h\|_\infty$.

(d) **($\mu$-Fisher IPM)** $\mathcal{F} = \left\{h : \mathbb{E}_{\mu(X)}\left[h^2(X)\right] \leq 1\right\}$ for some $\mu \in \mathscr{P}(\Omega)$ [33]. The penalty is $\Theta_{\mathcal{F}}(h) = \sqrt{\mathbb{E}_{\mu(X)}\left[h^2(X)\right]}$, however we can solve the infimum in Corollary 1 to get $\inf_{b \in \mathbb{R}} \Theta_{\mathcal{F}}(h - b) = \sqrt{\mathrm{Var}_\mu(h)}$ (Lemma 9 in Supplementary). This is interesting since the variance of $h$ as a penalty has appeared in work studying $f$-divergence uncertainty sets. Note that when $\mu = (P + Q)/2$ for some $P, Q \in \mathscr{P}(\Omega)$ then $d_{\mathcal{F}}(P, Q)$ is related to the $\chi^2$-divergence, the central $f$-divergence in these lines of work. In this setting, Corollary 1 extends the interpretation of variance regularization as a robustification strategy for any $\mu \in \mathscr{P}(\Omega)$.

Another interesting choice of $\mathcal{F}$ is the $\mu$-Sobelov IPM which we show in Table 1, whereby the resulting penalty is similar to those existing in manifold regularization [4]. All IPMs considered so far are of the form $\{h : \zeta(h) \leq 1\}$ for some $\zeta : \mathscr{F}(\Omega, \mathbb{R}) \to [0, \infty]$, and the resulting $\Theta_{\mathcal{F}}(h)$ closely resembles $\zeta(h)$. We derive $\Theta_{\mathcal{F}}$ for this general form with some assumptions on $\zeta$.

**Lemma 1** *Let $\zeta : \mathscr{F}(\Omega, \mathbb{R}) \to [0, \infty]$ be such that for some $k > 0$, $\zeta(a \cdot h) = a^k \cdot \zeta(h)$ for any $h \in \mathscr{F}(\Omega, \mathbb{R}), a > 0$. If $\mathcal{F} = \{h : \zeta(h) \leq 1\}$, then $\Theta_{\mathcal{F}}(h) \leq \sqrt[k]{\zeta(h)}$ with equality if $\zeta$ is convex.*

Our examples presented in Table 1 have convex choices of $\zeta$ with either $k = 1$ or $k = 2$. Using this Lemma, we may also interpret the case of two penalties added together, such as the Dudley metric in Table 1. Furthermore, Lemma 1 can be used for future applications of our work to elucidate robustness perspectives of methods using penalties of the form $\sqrt[k]{\zeta(h)}$.

We now return to the discussion on how closely related $\Lambda_{\mathcal{F},\varepsilon}$ is to $\varepsilon\Theta_{\mathcal{F}}$. Consider now two decompsitions of $h$ for the infimal convolution: $h_1 = 0, h_2 = h$ and $h_1 = h, h_2 = 0$, so we have $\Lambda_{\mathcal{F},\varepsilon}(h) \leq \varepsilon\Theta_{\mathcal{F}}(h)$ and $\Lambda_{\mathcal{F},\varepsilon}(h) \leq J_P(h)$ respectively. This yields $\Lambda_{\mathcal{F},\varepsilon}(h) \leq \min\left(J_P(h), \varepsilon\Theta_{\mathcal{F}}(h)\right)$, and we illustrate the tightness of this inequality through the following lemma.

**Lemma 2** *The mapping $h \mapsto \Lambda_{\mathcal{F},\varepsilon}(h)$ is subadditive and $\Lambda_{\mathcal{F},\varepsilon}(h)$ is the largest subadditive function that minorizes $\min\left(J_P(h), \varepsilon\Theta_{\mathcal{F}}(h)\right)$.*

The consequence of Lemma 2 is that if $\min\left(J_P(h), \varepsilon\Theta_{\mathcal{F}}(h)\right)$ is subadditive then $\Lambda_{\mathcal{F},\varepsilon}(h) = \min\left(J_P(h), \varepsilon\Theta_{\mathcal{F}}(h)\right)$ since a function always minorizes itself. In the proof of Lemma 2, we show that both $J_P$ and $\varepsilon\Theta_{\mathcal{F}}$ are subadditive and so if $\min\left(J_P, \varepsilon\Theta_{\mathcal{F}}\right)$ is consistently equal to either $J_P$ or $\varepsilon\Theta_{\mathcal{F}}$ for some $\varepsilon$ then we have equality.

We now present a necessary and sufficient condition for a function $h : \Omega \to \mathbb{R}$ so that $\Lambda_{\mathcal{F},\varepsilon}(h) = \varepsilon\Theta_{\mathcal{F}}(h)$ for all $\varepsilon > 0$. In doing so, not only do we lead to a better understanding of distributional robustness, we also contribute to understanding tightness of previous results and inequalities subsumed by Corollary 1. It turns out rather surprisingly that the characterization is directly related to penalty-regularized critic losses.

**Theorem 2** *A function $h \in \mathscr{F}(\Omega, \mathbb{R})$ satisfies $\Lambda_{\mathcal{F},\varepsilon}(h) = \Theta_{\mathcal{F}}(h)$ if and only if*

$$h \in \underset{\hat{h} \in \mathscr{F}(\Omega, \mathbb{R})}{\arg\inf} \left(\mathbb{E}_P[\hat{h}] - \mathbb{E}_\mu[\hat{h}] + \varepsilon\Theta_{\mathcal{F}}(\hat{h})\right), \tag{2}$$

*for some $\mu \in \mathscr{P}(\Omega)$.*

First, note that this characterization holds for any $h$ as long as one can find a $\mu$ that satisfies Equation (2). In particular, when $\mu = P$, then the minimizers of Equation (2) are constant functions. Furthermore, Equation (2) can be viewed as a regularized binary classification objective in the following way: $\Omega$ is the input space, $Y = \{-1, +1\}$ is the label space, $\hat{h} : \Omega \to \mathbb{R}$ is the classifier, $\Theta_{\mathcal{F}}$ is a penalty with weight $\varepsilon$, and $P$ (resp. $\mu$) corresponds to the $-1$ (resp. $+1$) class conditional

distribution. In particular, this is precisely the objective for the discriminator in penalty-based GANs [23, 51], referred to as the critic loss where $P$ is the fake data generated by a model and $\mu$ is the real data. Intuitively, the discriminator function will assign negative values to regions of $\mu$ and positive values to regions of $P$. The discriminator function is then used to guide learning of the model generator by focusing on moving $\mu$ to where $h$ assigns higher values. In conjunction with Theorem 1, this discriminator is robust to shifts to the distribution $P$ and we outline the consequence more clearly in the following Corollary.

**Corollary 2** *Let $P_+, P_- \in \mathscr{P}(\Omega)$ and suppose $\mathcal{F} \subseteq \mathscr{F}(\Omega, \mathbb{R})$ is even. If*

$$h^* \in \underset{\hat{h} \in \mathscr{F}(\Omega, \mathbb{R})}{\arg\inf} \left( \mathbb{E}_{P_-}[\hat{h}] - \mathbb{E}_{P_+}[\hat{h}] + \varepsilon \Theta_{\mathcal{F}}(\hat{h}) \right), \tag{3}$$

*then we have*

$$\inf_{Q \in B_{\varepsilon, \mathcal{F}}(P_+)} \int_\Omega h^* dQ = \int_\Omega h^* dP_+ - \varepsilon \Theta_{\mathcal{F}}(h^*)$$

$$\sup_{Q \in B_{\varepsilon, \mathcal{F}}(P_-)} \int_\Omega h^* dQ = \int_\Omega h^* dP_- + \varepsilon \Theta_{\mathcal{F}}(h^*).$$

The implication of this corollary is that the classifier learned by solving Equation (3) is still positive (resp. negative) around $B_{\varepsilon, \mathcal{F}}$ neighborhoods of $P_+$ (resp. $P_-$). In the context of GANs, $P_+$ and $P_-$ will be the real and fake distributions. This is a rather intuitive result since the classifier $h^*$ is penalized against $\Theta_{\mathcal{F}}$ however the above Corollary gives formal perspectives along with interpretations to the weighting $\varepsilon$ and the choice of penalty (induced by $\mathcal{F}$). We write this Corollary in a more general form since we believe it can be useful for other studies of robustness. An example of this is robustness certification: Given a binary classifier and reference distribution $\rho$, one can compute $\mathbb{E}_{\rho(X)}[h(X)] - \varepsilon \Theta_{\mathcal{F}}(-h)$ and check if this value is $\geq 0$. Using Definition 2.2 of [14] and Corollary 1 of our work, if this value is $\geq 0$ then this certifies that the classifier is robust to $\mathcal{F}$-IPM perturbations around $\rho$. This follows from the fact that Corollary 1 (using $-h$) implies $\mathbb{E}_{\rho(X)}[h(X)] - \varepsilon \Theta_{\mathcal{F}}(-h) \leq \inf_{Q \in B_{\varepsilon, \mathcal{F}}(\rho)} \mathbb{E}_{Q(X)}[h(X)]$ and positivity of the term on the right is precisely the condition laid out in Definition 2.2 of [15]. Corollary 2 uses the fact that the condition outlined in Theorem 2 is sufficient; however, we emphasize that it is also necessary, suggesting an intimate link between regularized binary-classification and distributional robustness.

# 4 Distributional Robustness of $f$-GANs

In this section, we show how our main theorem can naturally be applied into the robustness for $f$-GANs more generally. This is particularly relevant for the robustness community since as mentioned in the introduction, GANs are implemented as a robustifying mechanism for training binary classifiers. In this setting, $\Omega$ will typically be a high dimensional Euclidean space to represent the set of images and $P \in \mathscr{P}(\Omega)$ will be an empirical distribution that we are interested in modelling. The model distribution, also referred to as the generative distribution denoted as $\mu \in \mathscr{P}(\Omega)$, is learned by minimizing a divergence between $P$ and $\mu$. We now introduce the $f$-GAN objective, which is a central divergence in the GAN paradigm.

**Definition 4 ($f$-GAN, [37])** *Let $f : \mathbb{R} \to (-\infty, \infty]$ be a lower semicontinuous convex function with $f(1) = 0$ and $\mathcal{H} \subset \mathscr{F}(\Omega, \mathrm{dom}\, f^\star)$ be a set of discriminators. The GAN objective for data $P \in \mathscr{P}(\Omega)$ and model $\mu \in \mathscr{P}(\Omega)$ is*

$$\mathrm{GAN}_{f,\mathcal{H}}(\mu; P) = \sup_{h \in \mathcal{H}} \left( \int_\Omega h \, dP - \int_\Omega f^\star(h) \, d\mu \right),$$

*where $f^\star(y) = \sup_{x \in \mathbb{R}} (x \cdot y - f(x))$ is the convex conjugate.*

We are interested in minimizing the above objective with respect to $\mu$, which results in a min-max objective due to the supremum taken over $\mathcal{H}$. One should note that there are two components of this objective that characterize it, the function $f$ and discriminator set $\mathcal{H}$. In practice, the discriminator set is often restricted, and so the resulting objective is not a divergence; however, empirical studies

have observed convergence [19], which warrants an investigation into the effects of a restricted discriminator on model performance. Existing theoretical work has hinted the benefits of a restricted discriminator, for example, [56] show that generalization is related to the Rademacher complexity of the discriminator set and suggest a discrimination-generalization trade-off. Other work has suggested that the particular setting of Lipschitz discriminators leads to improvements for both practical [56, 19, 59, 54, 18] and theoretical purposes [25, 18, 30]. It is clear that the discriminator set is a key character in the tale of success of GANs; however, the existing literature is silent on what it means for robustness, a particular application that GANs have posed successful in, and this is precisely the link we establish with the following Theorem.

**Theorem 3** *Let* $f : \mathbb{R} \rightarrow \mathbb{R}$ *be a convex lower semi-continuous function with* $f(1) = 0$, $\mathcal{F} \subseteq \mathscr{F}(\Omega, \mathbb{R})$ *and* $\mathcal{H} \subseteq \mathscr{F}(\Omega, \mathrm{dom}(f^\star))$. *For any model and data distributions* $\mu, P \in \mathscr{P}(\Omega)$ *respectively, we have for all* $\varepsilon > 0$

$$\sup_{Q \in B_{\varepsilon, \mathcal{F}}(P)} \mathrm{GAN}_{f, \mathcal{H}}(\mu; Q) \leq \mathrm{GAN}_{f, \mathcal{H}}(\mu; P) + \varepsilon \sup_{h \in \mathcal{H}} \Theta_{\mathcal{F}}(h).$$

This Theorem tells us that the robust version of the GAN objective can be upper bounded by the standard GAN objective plus a term that quantifies the complexity of the discriminator set. Note that the robustness parameters ($\varepsilon$ and $\mathcal{F}$) interact only with the discriminator set and not the generative model $\mu$, revealing the importance of choosing a regularized discriminator set $\mathcal{H}$. To see this more clearly, consider the setting $\mathcal{F} = \mathcal{H}$, and since $\Theta_{\mathcal{H}}(h) \leq 1$, we have

$$\sup_{Q \in B_{\varepsilon, \mathcal{H}}(P)} \mathrm{GAN}_{f, \mathcal{H}}(\mu; Q) \leq \mathrm{GAN}_{f, \mathcal{H}}(\mu; P) + \varepsilon, \tag{4}$$

for all $\varepsilon > 0$. The key insight is that training GANs using discriminators $\mathcal{H}$ yields guarantees on the robust GAN objective for adversaries who pick $Q$ from $B_{\varepsilon, \mathcal{H}}(P)$. Note that if one picks discriminators $\mathcal{H}$ that are too strong then the ball $B_{\varepsilon, \mathcal{H}}(P)$ will shrink and become singleton $\{P\}$ when $\mathcal{H} = \mathscr{F}(\Omega, \mathbb{R})$. On the other hand, if $\mathcal{H}$ is chosen to be smaller then the uncertainty set is larger; however, the first term $\mathrm{GAN}_{f, \mathcal{H}}$ will be a weaker divergence, since the discriminator set determines the strength of the objective [30]. Hence, there is a trade-off between discrimination and robustness, that complements and parallels the discrimination-generalization story described in [56].

We now discuss the particular settings of $\mathcal{F}$ and how our theorem gives a perspective of distributional robustness on existing GAN methods. First, consider choices of $\mathcal{F}$ so that $d_{\mathcal{F}}$ corresponds to MMD, Fisher IPM and Sobelov IPM which translates to the MMD-GAN, Fisher-GAN and Sobelov GAN respectively, allowing us to view these methods from a robustness perspective in light of Theorem 3 and Equation (4). Furthermore, our result also contributes to the positive commentary under the popular choice of Lipschitz regularized discriminators, guarantees against adversaries selecting from Wasserstein uncertainty sets. It should be noted that recently, a method that regularizes discriminators by minimizing a penalty referred to as 0-GP [51] has proven convergence and generalization guarantees. It can be easily shown that this penalty satisfies the conditions of Lemma 1 for $k = 2$ due to its resemblance to the Sobelov IPM, allowing us to present a robustness interpretation for this penalty. Our main insight from this perspective reveals the theoretical benefits of regularized discriminators. In light of our results, learning a binary classifier using a GAN (trained with regularized discriminators) as a downstream task implies this classifier will consequently be robust.

## 5 Conclusion

Our results extend the Distributionally Robust Optimization (DRO) framework to IPMs, which reveal further importance of the role regularization plays for robustness and machine learning at large. Unlike most DRO applications to machine learning, we present equality and show that achieving this is fundamentally rooted in regularized binary classification. We then show that DRO can be extended to understand GANs and unveil the role of discrimination regularization in these frameworks. The results will also help DRO explain regularization penalties through the lens of robustness in the future. Our contributions are modular and pave the way to build on related areas, one such example being robustness certification, which we leave for the subject of future work.

## Broader Impact

From the perspective of impact, the main contribution of our work is understanding how regularization, a commonly used technique in machine training, gives benefits for robustness. We show this for different areas of machine learning, such as supervised learning and generative adversarial networks. The ultimate goal of such work is to develop further our understanding of these methods and how their performances can be improved. Our work does not have a focused application use-case under which we can discuss specific ethical considerations since it contributes more generally to the advancements of performance. In this sense, ethical considerations are subject to the application of these methods.

## Acknowledgments and Disclosure of Funding

We would like to thank Jeremias Knoblauch and anonymous reviewers for comments regarding the focus and clarity of presentation. This work was funded by the Australian Government Research Training Program and Data61.

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
