[Supplementary Material]

# Supplementary Material for "Distributional Robustness with IPMs and links to Regularization and GANs"

**Hisham Husain**
The Australian National University & Data61
`hisham.husain@anu.edu.au`

## Abstract

This document contains the proofs for the main results of the submission "Distributional Robustness with IPMs and links to Regularization and GANs".

## Appendix: Table of Contents

# 1 Proofs of Main Results

Before we begin, we introduce some notation that will be used to prove the main results that is exclusive to the Appendix. We will be invoking general convex analysis on the space $\mathscr{F}(\Omega, \mathbb{R})$, in the same fashion as [2], noting that $\mathscr{F}(\Omega, \mathbb{R})$ is a Hausdorff locally convex space (through the uniform norm). We use $\mathscr{B}(\Omega)$ to denote the denote the set of all bounded and finitely additive signed measures over $\Omega$ (with a given $\sigma$-algebra). For any set $D \subseteq \mathscr{B}(\Omega)$ and $h \in \mathscr{F}(\Omega, \mathbb{R})$, we use $\sigma_D(h) = \sup_{\nu \in D} \langle h, \nu \rangle$ and $\delta_D(\nu) = \infty \cdot [\![\nu \notin D]\!]$ to denote the *support* and *indicator* functions such as in [5]. We introduce the conjugate specific to these spaces

**Definition 1 ([6])** *For any proper convex function $F : \mathscr{F}(\Omega, \mathbb{R}) \to (-\infty, \infty)$, we have for any $\mu \in \mathscr{B}(\Omega)$ we define*

$$F^\star(\mu) = \sup_{h \in \mathscr{F}(\Omega, \mathbb{R})} \left( \int_\Omega h d\mu - F(h) \right)$$

*and for any $h \in \mathscr{F}(\Omega, \mathbb{R})$ we define*

$$F^{\star\star}(h) = \sup_{\mu \in \mathscr{B}(\Omega)} \left( \int_\Omega h d\mu - F^\star(\mu) \right).$$

**Theorem 1 ([8] Theorem 2.3.3)** *If $X$ is a Hausdorff locally convex space, and $F : X \to (-\infty, \infty]$ is a proper convex lower semi-continuous function then $F^{\star\star} = F$.*

There is an additional robustness result which we will deploying for several proofs which holds for any space $A$ that admits Polish topology.

**Lemma 1** *For any $\mathcal{F} \subseteq \mathscr{F}(\Omega, \mathbb{R})$, we have that*

$$d_\mathcal{F}(P, \mu) = d_{\overline{\mathrm{co}}(\mathcal{F})}(P, \mu).$$

**Proof** Let $\Delta_n := \{\alpha \in [0, 1]^n : \sum_{i=1}^n \alpha = 1\}$ Note that we have

$$
\begin{aligned}
d_{\mathrm{co}(\mathcal{F})}(P, \mu) &= \sup_{n \in \mathbb{N}, \alpha \in \Delta_n, f_i \in \mathcal{F} \forall i = 1, \ldots, n} \left\{ \mathbb{E}_P \left[ \sum_{i=1}^n \alpha_i f_i \right] - \mathbb{E}_\mu \left[ \sum_{i=1}^n \alpha_i f_i \right] \right\} \\
&= \sup_{n \in \mathbb{N}, \alpha \in \Delta_n, f_i \in \mathcal{F} \forall i = 1, \ldots, n} \sum_{i=1}^n \alpha_i \{ \mathbb{E}_P[f_i] - \mathbb{E}_\mu[f_i] \} \\
&= \sup_{n \in \mathbb{N}, \alpha \in \Delta_n} \sum_{i=1}^n \alpha_i \sup_{f_i \in \mathcal{F}} \{ \mathbb{E}_P[f_i] - \mathbb{E}_\mu[f_i] \} \\
&= \sup_{n \in \mathbb{N}, \alpha \in \Delta_n} \sum_{i=1}^n \alpha_i d_\mathcal{F}(P, \mu) \\
&= d_\mathcal{F}(P, \mu)
\end{aligned}
$$

It is also closed under taking the closure since $d_\mathcal{F}$ is the supremum of continuous (linear) functions and the supremum over a set with a linear objective is equal to taking the supremum over the closure of that set. ∎

**Definition 2** *For any $\mathcal{F} \subseteq \mathscr{F}(\Omega, \mathbb{R})$, we define the functional $R_\mathcal{F} : \mathscr{F}(\Omega, \mathbb{R}) \to [0, \infty]$ as*

$$R_\mathcal{F}(h) := \int_\Omega h dP + \delta_{\overline{\mathrm{co}}(\mathcal{F})}(h).$$

**Lemma 2** *For any $\mathcal{F} \subseteq \mathscr{F}(\Omega, \mathbb{R})$, $R_\mathcal{F}$ is proper convex and lower semi-continuous.*

**Proof** The mapping $h \mapsto \int_\Omega h dP$ is clearly convex and lower semi-continuous. Since $\overline{\text{co}}\,(\mathcal{F})$ is a closed and convex set, the indicator function $\delta_{\overline{\text{co}}(\mathcal{F})}(h)$ is proper convex and lower semi-continuous and thus the result follows. ∎

**Lemma 3** *The mappings $\nu \mapsto d_{\mathcal{F}}(\nu, P)$ and $h \mapsto R_{\mathcal{F}}(h)$ are convex conjugates*

**Proof** Note first that for any $\nu \in \mathscr{B}(\Omega)$

$$R_{\mathcal{F}}^\star(\nu) = \sup_{h \in \mathscr{F}(\Omega, \mathbb{R})} \left\{ \int_\Omega h d\nu - \int_\Omega h dP - \delta_{\overline{\text{co}}(\mathcal{F})}(h) \right\}$$

$$= \sup_{h \in \overline{\text{co}}(\mathcal{F})} \left\{ \int_\Omega h d\nu - \int_\Omega h dP \right\}$$

$$= d_{\overline{\text{co}}(\mathcal{F})}(\nu, P)$$

$$\overset{(1)}{=} d_{\mathcal{F}}(\nu, P),$$

where $(1)$ is due to Lemma 1. We also have that

$$(d_{\mathcal{F}}(\cdot, P))^\star (h) = \sup_{\nu \in \mathscr{B}(\Omega)} \left\{ \int_\Omega h d\nu - d_{\mathcal{F}}(\nu, P) \right\}$$

$$\overset{(1)}{=} \sup_{\nu \in \mathscr{B}(\Omega)} \left\{ \int_\Omega h d\nu - R_{\mathcal{F}}^\star(\nu) \right\}$$

$$\overset{(2)}{=} R_{\mathcal{F}}^{\star\star}(\nu)$$

$$\overset{(3)}{=} R_{\mathcal{F}}(\nu),$$

where $(1)$ holds due to the above, $(2)$ holds by definition of conjugate and $(3)$ holds by a combination of Lemma 1 and Lemma 2. ∎

We also present a lemma which will prove to be useful in proving the main results.

**Lemma 4** *For any $\mathcal{F} \subset \mathscr{F}(\Omega, \mathbb{R})$, the mapping $h \mapsto \Theta_{\mathcal{F}}(h)$ is convex.*

**Proof** First notice that for any $t > 0$ and $h \in \mathscr{F}(\Omega, \mathbb{R})$ we have that $\Theta_{\mathcal{F}}(t \cdot h) = t \cdot \Theta_{\mathcal{F}}(h)$. For any $t \in [0, 1]$ and $h, h' \in \mathscr{F}(\Omega, \mathbb{R})$, consider the element $\tilde{h} := t \cdot h + (1 - t) \cdot h'$. Since $t \cdot h \in t\Theta_{\mathcal{F}}(h) \cdot \overline{\text{co}}\,(\mathcal{F})$ and $(1 - t)h \in (1 - t)\Theta_{\mathcal{F}}(h) \cdot \overline{\text{co}}\,(\mathcal{F})$, we have that

$$\tilde{h} \in t\Theta_{\mathcal{F}}(h) \cdot \overline{\text{co}}\,(\mathcal{F}) + (1 - t)\Theta_{\mathcal{F}}(h) \cdot \overline{\text{co}}\,(\mathcal{F})$$

$$\iff \tilde{h} \in (t\Theta_{\mathcal{F}}(h) + (1 - t)\Theta_{\mathcal{F}}(h')) \cdot \overline{\text{co}}\,(\mathcal{F}),$$

which in turn implies that $\Theta_{\mathcal{F}}(\tilde{h}) \leq t\Theta_{\mathcal{F}}(h) + (1 - t)\Theta_{\mathcal{F}}(h')$, proving convexity of $\Theta_{\mathcal{F}}$. ∎

## 1.1 Proof of Theorem 1

**Theorem 2** *Let $\mathcal{F} \subseteq \mathscr{F}(\Omega, \mathbb{R})$ and $P \in \mathscr{P}(\Omega)$. For any $h \in \mathscr{F}(\Omega, \mathbb{R})$ and for all $\varepsilon > 0$*

$$\sup_{Q \in B_{\varepsilon, \mathcal{F}}(P)} \int_\Omega h dQ = \int_\Omega h dP + \Lambda_{\mathcal{F}, \varepsilon}(h).$$

**Proof** We first require two lemmata.

**Lemma 5** *For any $\mathcal{F} \subseteq \mathscr{F}(\Omega, \mathbb{R})$, $P \in \mathscr{P}(\Omega)$, $\lambda \geq 0$ and $h \in \mathscr{F}(\Omega, \mathbb{R})$, we have*

$$\sup_{Q \in \mathscr{P}(\Omega)} \left( \int_\Omega h dQ - \lambda d_{\mathcal{F}}(Q, P) \right) = R_{\lambda \mathcal{F}} \overline{\star} \sigma_{\mathscr{P}(\Omega)}(h)$$

**Proof** We use a standard result from convex analysis which states that the convex conjugate of the sum of two functions is the infimal convolution of their conjugates. Hence we have

$$
\sup_{Q \in \mathscr{P}(\Omega)} \left( \int_\Omega h dQ - \lambda d_{\mathcal{F}}(Q,P) \right) = \sup_{Q \in \mathscr{B}(\Omega)} \left( \int_\Omega h dQ - \lambda d_{\mathcal{F}}(Q,P) - \delta_{\mathscr{P}(\Omega)}(Q) \right)
$$

$$
= \left( \lambda d_{\mathcal{F}}(Q,P) + \delta_{\mathscr{P}(\Omega)}(Q) \right)^\star
$$

$$
= \left( \lambda d_{\mathcal{F}}(Q,P) \right)^\star \mathbin{\overline{\star}} \left( \delta_{\mathscr{P}(\Omega)}(Q) \right)^\star
$$

$$
= R_{\lambda \mathcal{F}} \mathbin{\overline{\star}} \sigma_{\mathscr{P}(\Omega)}(h),
$$

which follows from Lemma 3 and the fact that support functions are conjugates of indicator functions [4, Section 3.4.1, Example (a)]. ∎

**Lemma 6** *For any $\mathcal{F} \subseteq \mathscr{F}(\Omega, \mathbb{R})$, $P \in \mathscr{P}(\Omega)$, and $h \in \mathscr{F}(\Omega, \mathbb{R})$, we have*

$$
\inf_{\lambda \geq 0} \left( R_{\lambda \mathcal{F}} \mathbin{\overline{\star}} \sigma_{\mathscr{P}(\Omega)}(h) + \lambda \varepsilon \right) = \int_\Omega h dP + J_P \mathbin{\overline{\star}} \varepsilon \Theta_{\mathcal{F}}(h)
$$

**Proof** Using the definition of infimal convolution, we have

$$
\inf_{\lambda \geq 0} \left( R_{\lambda \mathcal{F}} \mathbin{\overline{\star}} \sigma_{\mathscr{P}(\Omega)}(h) + \lambda \varepsilon \right)
$$

$$
= \inf_{\lambda \geq 0} \left( \inf_{h' \in \mathscr{F}(\Omega, \mathbb{R})} \left( \int_\Omega (h - h') dP + \delta_{\overline{\mathrm{co}}(\lambda \mathcal{F})}(h - h') + \sigma_{\mathscr{P}(\Omega)}(h) \right) + \lambda \varepsilon \right)
$$

$$
= \inf_{\lambda \geq 0} \inf_{h' \in \mathscr{F}(\Omega, \mathbb{R})} \left( \int_\Omega h dP - \int_\Omega h' dP + \delta_{\overline{\mathrm{co}}(\lambda \mathcal{F})}(h - h') + \sigma_{\mathscr{P}(\Omega)}(h') + \lambda \varepsilon \right)
$$

$$
= \int_\Omega h dP + \inf_{h' \in \mathscr{F}(\Omega, \mathbb{R})} \left( - \int_\Omega h' dP + \inf_{\lambda \geq 0} \left( \delta_{\overline{\mathrm{co}}(\lambda \mathcal{F})}(h - h') + \lambda \varepsilon \right) + \sigma_{\mathscr{P}(\Omega)}(h') \right)
$$

$$
= \int_\Omega h dP + \inf_{h' \in \mathscr{F}(\Omega, \mathbb{R})} \left( \sigma_{\mathscr{P}(\Omega)}(h') - \int_\Omega h' dP + \inf_{\lambda \geq 0} \left( \delta_{\overline{\mathrm{co}}(\lambda \mathcal{F})}(h - h') + \lambda \varepsilon \right) \right)
$$

$$
= \int_\Omega h dP + \inf_{h' \in \mathscr{F}(\Omega, \mathbb{R})} \left( \sigma_{\mathscr{P}(\Omega)}(h') - \int_\Omega h' dP + \inf_{\lambda \geq 0} \left( \infty \cdot [\![ h - h' \notin \lambda \cdot \overline{\mathrm{co}}(\mathcal{F}) ]\!] + \lambda \varepsilon \right) \right)
$$

$$
= \int_\Omega h dP + \inf_{h' \in \mathscr{F}(\Omega, \mathbb{R})} \left( J_P(h') + \varepsilon \Theta_{\mathcal{F}}(h - h') \right)
$$

$$
= \int_\Omega h dP + J_P \mathbin{\overline{\star}} \varepsilon \Theta_{\mathcal{F}}(h).
$$

∎

We are now ready to prove the Theorem. By introducing a dual variable $\lambda > 0$ that penalizes the ball constraint, we have

$$
\sup_{Q \in B_{\varepsilon, \mathcal{F}}(P)} \int_\Omega h dQ = \sup_{Q \in \mathscr{P}(\Omega) : d_{\mathcal{F}}(Q,P) \leq \varepsilon} \int_\Omega h dQ
$$

$$
= \sup_{Q \in \mathscr{P}(\Omega)} \inf_{\lambda \geq 0} \left( \int_\Omega h dQ + \lambda (\varepsilon - d_{\mathcal{F}}(Q,P)) \right)
$$

$$
\overset{(1)}{=} \inf_{\lambda \geq 0} \sup_{Q \in \mathscr{P}(\Omega)} \left( \int_\Omega h dQ + \lambda (\varepsilon - d_{\mathcal{F}}(Q,P)) \right)
$$

$$
= \inf_{\lambda \geq 0} \left( \sup_{Q \in \mathscr{P}(\Omega)} \left( \int_\Omega h dQ - \lambda d_{\mathcal{F}}(Q,P) \right) + \lambda \varepsilon \right)
$$

$$
\overset{(2)}{=} \inf_{\lambda \geq 0} \left( R_{\lambda \mathcal{F}} \mathbin{\overline{\star}} \sigma_{\mathscr{P}(\Omega)}(h) + \lambda \varepsilon \right)
$$

$$
\overset{(3)}{=} \int_\Omega h dP + J_P \mathbin{\overline{\star}} \varepsilon \Theta_{\mathcal{F}}(h),
$$

where (2) and (3) hold due to Lemma 5 and 6 respectively. To see why (1) holds, first note that the mapping $Q \mapsto \int_\Omega hdQ + \lambda \left( \varepsilon - d_\mathcal{F}(Q, P) \right)$ is concave and lower semicontinuous since $d_\mathcal{F}$ is the supremum of linear functions. Next we have by an application of the Banach-Alaogu Theorem that $\mathscr{P}(\Omega)$ is compact [2, Lemma 27 (b)]. Hence by [1, Theorem 2], (1) follows. $\blacksquare$

## 1.2 Proof of Corollary 1

**Corollary 1** *Let* $\mathcal{F} \subseteq \mathscr{F}(\Omega, \mathbb{R})$ *and* $P \in \mathscr{P}(\Omega)$. *For any* $h \in \mathscr{F}(\Omega, \mathbb{R})$ *and for all* $\varepsilon > 0$

$$\sup_{Q \in B_{\varepsilon, \mathcal{F}}(P)} \int_\Omega hdQ \leq \int_\Omega hdP + \varepsilon \inf_{b \in \mathbb{R}} \Theta_\mathcal{F}(h - b).$$

**Proof** By definition of the infimal convolution we can consider a decomposition of the form $h_1 = b$ and $h_2 = h - b$ for some $b \in \mathbb{R}$. notice that $J_P(b) = 0$ and by taking the smallest possible $b \in \mathbb{R}$ yields

$$\Theta_{\mathcal{F}, \varepsilon}(h) \leq \varepsilon \inf_{b \in \mathbb{R}} \Theta_\mathcal{F}(h - b),$$

which completes the proof. $\blacksquare$

## 1.3 Proof of Lemma 1

**Lemma 7** *Let* $\zeta : \mathscr{F}(\Omega, \mathbb{R}) \to [0, \infty]$ *be a penalty such that* $\zeta(a \cdot h) = a^k \cdot \zeta(h)$ *for any* $h \in \mathscr{F}(\Omega, \mathbb{R})$, $k, a > 0$. *Let* $\mathcal{F} = \{h : \zeta(h) \leq 1\}$ *then we have* $\Theta_\mathcal{F}(h) \leq \sqrt[k]{\zeta(h)}$ *with equality if* $\zeta$ *is convex.*

**Proof** Let us consider the non-convex case so that $\mathcal{F}$ is not necessarily convex. We then have for any $\mathcal{F} \subseteq \mathscr{F}(\Omega, \mathbb{R})$

$$h \in \overline{\mathrm{co}}\left( \lambda \mathcal{F} \right) \iff h \in \lambda \overline{\mathrm{co}}\left( \mathcal{F} \right)$$
$$\iff \frac{h}{\lambda} \in \overline{\mathrm{co}}\left( \mathcal{F} \right)$$

For a fixed $h \in \mathscr{F}(\Omega, \mathbb{R})$, set $\lambda = \sqrt[k]{\zeta(h)}$ and notice that

$$\zeta\left( \frac{h}{\lambda} \right) = \zeta\left( \frac{h}{\sqrt[k]{\zeta(h)}} \right)$$
$$= \left( \frac{1}{\sqrt[k]{\zeta(h)}} \right)^k \zeta(h)$$
$$= \zeta(h),$$

and so we have $\Theta_\mathcal{F}(h) \leq \sqrt[k]{\zeta(h)}$. In the case when the penalty is convex, we have that $\mathcal{F}$ will be convex and so

$$h \in \lambda \overline{\mathrm{co}}\left( \mathcal{F} \right) \iff \frac{h}{\lambda} \in \overline{\mathrm{co}}\left( \mathcal{F} \right)$$
$$\iff \frac{h}{\lambda} \in \mathcal{F}$$
$$\iff \zeta\left( \frac{h}{\lambda} \right) \leq 1$$
$$\iff \frac{1}{\lambda^k} \zeta(h) \leq 1$$
$$\iff \zeta(h) \leq \lambda^k$$
$$\iff \sqrt[k]{\zeta(h)} \leq \lambda.$$

Hence we have $\Theta_\mathcal{F}(h) = \inf_{\sqrt[k]{\zeta(h)} \leq \lambda} \lambda = \sqrt[k]{\zeta(h)}$. $\blacksquare$

### 1.4 Proof of Lemma 2

**Lemma 8** *The mapping $h \mapsto \Lambda_{\mathcal{F},\varepsilon}(h)$ is subadditive and $\Lambda_{\mathcal{F},\varepsilon}(h)$ is the largest subadditive function that minorizes $\min\left(J_P(h), \varepsilon\Theta_{\mathcal{F}}(h)\right)$.*

**Proof** Since $\Theta_{\mathcal{F}}(h)$ is convex (Lemma 4) and $\Theta_{\mathcal{F}}(t \cdot h) = t \cdot \Theta_{\mathcal{F}}(h)$ for $t > 0$, it follows that $\Theta_{\mathcal{F}}(h)$ is subadditive. Next notice that $J_P$ is subadditive since for any $h, h' \in \mathscr{F}(\Omega, \mathbb{R})$

$$
\begin{aligned}
J_P(h + h') &= \sup_{\omega \in \Omega} h(\omega) + h'(\omega) - \int_{\Omega} h \, dP - \int_{\Omega} h' \, dP \\
&\leq \sup_{\omega \in \Omega} h(\omega) - \int_{\Omega} h \, dP + \sup_{\omega \in \Omega} h'(\omega) - \int_{\Omega} h' \, dP \\
&= J_P(h) + J_P(h').
\end{aligned}
$$

Next notice that $J_P(0) = 0$ and $\varepsilon\Theta_{\mathcal{F}}(0) = 0$. By [7, Theorem 2.5(c)] we have that $\Lambda_{\mathcal{F},\varepsilon}$ is sub-additive and that it is the largest subadditive function that minorizes $\min\left(J_P(h), \varepsilon\Theta_{\mathcal{F}}(h)\right)$. ∎

### 1.5 Proof of Theorem 2

**Theorem 3** *A function $h \in \mathscr{F}(\Omega, \mathbb{R})$ satisfies $\Lambda_{\mathcal{F},\varepsilon}(h) = \Theta_{\mathcal{F}}(h)$ if and only if*

$$
h \in \operatorname*{arg\,inf}_{\hat{h} \in \mathscr{F}(\Omega, \mathbb{R})} \left( \mathbb{E}_P[\hat{h}] - \mathbb{E}_\mu[\hat{h}] + \varepsilon\Theta_{\mathcal{F}}(\hat{h}) \right),
$$

*for some $\mu \in \mathscr{P}(\Omega)$.*

**Proof** To prove this Theorem, we use the conditions for an optimal decomposition of an infimal convolution as shown in [3, Lemma 1]. First note that $J_P$ and $\Theta_{\mathcal{F}}$ are convex (Lemma 4). Note that the property is equivalent to showing that the decomposition $h_1 = 0$ and $h_2 = h$ is optimal. By [3, Lemma 1], this decomposition is optimal if and only if there exists a measure $\nu^* \in \mathscr{B}(\Omega)$ such that

$$
J_P(0) = \langle \nu^*, 0 \rangle - J_P^\star(\nu^*) \tag{1}
$$
$$
\varepsilon\Theta_{\mathcal{F}}(h) = \langle \nu^*, h \rangle - (\varepsilon\Theta_{\mathcal{F}})^\star(\nu^*) \tag{2}
$$

First note that $J_P(h) = \sigma_{\mathscr{P}(\Omega)}(h) + \sigma_{\{-P\}}(h)$ and using properties of infimal convolutions, we have for any $\nu \in \mathscr{P}(\Omega)$

$$
\begin{aligned}
J_P^\star(\nu) &= \left( \sigma_{\mathscr{P}(\Omega)} + \sigma_{\{-P\}} \right)^\star (\nu) \\
&= \left( \sigma_{\mathscr{P}(\Omega)}^\star \,\overline{\star}\, \sigma_{\{-P\}}^\star \right)(\nu) \\
&= \left( \delta_{\mathscr{P}(\Omega)} \,\overline{\star}\, \delta_{\{-P\}} \right)(\nu) \\
&= \inf_{\nu' \in \mathscr{B}(\Omega)} \left( \delta_{\mathscr{P}(\Omega)}(\nu') + \delta_{\{-P\}}(\nu - \nu') \right) \\
&= \inf_{\nu' \in \mathscr{P}(\Omega)} \delta_{\{-P\}}(\nu - \nu') \\
&= \infty \cdot [\![ P + \nu \notin \mathscr{P}(\Omega) ]\!] \\
&= \infty \cdot [\![ \nu \notin \mathscr{P}(\Omega) - P ]\!].
\end{aligned}
$$

Since $J_P(0) = \langle \nu, 0 \rangle = 0$ for any $\nu \in \mathscr{B}(\Omega)$, this tells us that a $\nu^*$ satisfies the condition of Equation 1 if and only if $\nu^*$ is of the form $\mu - P$ where $\mu$ is any element of $\mathscr{P}(\Omega)$. We can re-arrange Equation 2 into

$$
\langle \nu^*, h \rangle - \varepsilon\Theta_{\mathcal{F}}(h) = (\varepsilon\Theta_{\mathcal{F}})^\star(\nu^*),
$$

and by definition since $(\varepsilon\Theta_{\mathcal{F}})^{\star}(\nu^*) = \sup_{\hat{h}\in\mathscr{F}(\Omega,\mathbb{R})}\left(\left\langle \nu^*, \hat{h}\right\rangle - \varepsilon\Theta_{\mathcal{F}}(\hat{h})\right)$, Equation 2 setting $\nu^* = \mu - P$ becomes

$$\langle \nu^*, h\rangle - \varepsilon\Theta_{\mathcal{F}}(h) = \sup_{\hat{h}\in\mathscr{F}(\Omega,\mathbb{R})}\left(\left\langle \nu^*, \hat{h}\right\rangle - \varepsilon\Theta_{\mathcal{F}}(\hat{h})\right) \tag{3}$$

$$\iff \langle \mu - P, h\rangle - \varepsilon\Theta_{\mathcal{F}}(h) = \sup_{\hat{h}\in\mathscr{F}(\Omega,\mathbb{R})}\left(\left\langle \mu - P, \hat{h}\right\rangle - \varepsilon\Theta_{\mathcal{F}}(\hat{h})\right)$$

$$\iff \mathbb{E}_{\mu}[h] - \mathbb{E}_P[h] - \varepsilon\Theta_{\mathcal{F}}(h) = \sup_{\hat{h}\in\mathscr{F}(\Omega,\mathbb{R})}\left(\mathbb{E}_{\mu}[\hat{h}] - \mathbb{E}_P[\hat{h}] - \varepsilon\Theta_{\mathcal{F}}(\hat{h})\right)$$

$$\iff h \in \operatorname*{arg\,sup}_{\hat{h}\in\mathscr{F}(\Omega,\mathbb{R})}\left(\mathbb{E}_{\mu}[\hat{h}] - \mathbb{E}_P[\hat{h}] - \varepsilon\Theta_{\mathcal{F}}(\hat{h})\right)$$

$$\iff h \in \operatorname*{arg\,inf}_{\hat{h}\in\mathscr{F}(\Omega,\mathbb{R})}\left(\mathbb{E}_P[\hat{h}] - \mathbb{E}_{\mu}[\hat{h}] + \varepsilon\Theta_{\mathcal{F}}(\hat{h})\right). \tag{4}$$

Hence the decomposition $h_1 = 0$ and $h_2 = h$ is optimal if and only if $h$ satisfies Equation 4 for some $\mu \in \mathscr{P}(\Omega)$, which is precisely the statement of the Theorem. ∎

## 1.6 Proof of Corollary 2

**Corollary 2** *Let $P_+, P_- \in \mathscr{P}(\Omega)$ and suppose $\mathcal{F} \subseteq \mathscr{F}(\Omega,\mathbb{R})$ is even. If*

$$h^* \in \operatorname*{arg\,inf}_{\hat{h}\in\mathscr{F}(\Omega,\mathbb{R})}\left(\mathbb{E}_{P_-}[\hat{h}] - \mathbb{E}_{P_+}[\hat{h}] + \varepsilon\Theta_{\mathcal{F}}(\hat{h})\right),$$

*then we have*

$$\inf_{Q\in B_{\varepsilon,\mathcal{F}}(P_+)}\int_{\Omega} h^* dQ = \int_{\Omega} h^* dP_+ - \varepsilon\Theta_{\mathcal{F}}(h^*)$$

$$\sup_{Q\in B_{\varepsilon,\mathcal{F}}(P_-)}\int_{\Omega} h^* dQ = \int_{\Omega} h^* dP_- + \varepsilon\Theta_{\mathcal{F}}(h^*)$$

**Proof** Applying Theorem 2 with $P = P_-$ and $\mu = P_+$ and using Theorem 1 yields the result on $B_{\varepsilon,\mathcal{F}}(P_-)$. Notice that $\mathcal{F}$ is even, which means that $\Theta_{\mathcal{F}}(h) = \Theta_{\mathcal{F}}(-h)$ and so we have

$$h^* \in \operatorname*{arg\,inf}_{\hat{h}\in\mathscr{F}(\Omega,\mathbb{R})}\left(\mathbb{E}_{P_-}[\hat{h}] - \mathbb{E}_{P_+}[\hat{h}] + \varepsilon\Theta_{\mathcal{F}}(\hat{h})\right)$$

$$\iff -h^* \in \operatorname*{arg\,inf}_{-\hat{h}\in\mathscr{F}(\Omega,\mathbb{R})}\left(-\mathbb{E}_{P_-}[\hat{h}] + \mathbb{E}_{P_+}[\hat{h}] + \varepsilon\Theta_{\mathcal{F}}(-\hat{h})\right)$$

$$\iff -h^* \in \operatorname*{arg\,inf}_{-\hat{h}\in\mathscr{F}(\Omega,\mathbb{R})}\left(\mathbb{E}_{P_+}[\hat{h}] - \mathbb{E}_{P_-}[\hat{h}] + \varepsilon\Theta_{\mathcal{F}}(\hat{h})\right).$$

We can then apply Theorem 2 to $-h^*$ which means $\Lambda_{\varepsilon,\mathcal{F}}(-h^*) = \varepsilon\Theta_{\mathcal{F}}(-h^*) = \varepsilon\Theta_{\mathcal{F}}(h^*)$. Putting this together and applying Theorem 1 to $-h^*$ gives

$$\sup_{Q\in B_{\varepsilon,\mathcal{F}}(P_+)}\int_{\Omega} -h^* dQ = \int_{\Omega} -h^* dP_+ + \varepsilon\Theta_{\mathcal{F}}(h^*),$$

and multiplying both sides by $-1$ concludes the proof. ∎

## 1.7 Proof of Theorem 3

**Theorem 4** *Let $f : \mathbb{R} \to \mathbb{R}$ be a convex lower semi-continuous function with $f(1) = 0$, $\mathcal{F} \subseteq \mathscr{F}(\Omega,\mathbb{R})$ and $\mathcal{H} \subseteq \mathscr{F}(\Omega,\operatorname{dom}(f^{\star}))$. For any model and data distributions $\mu, P \in \mathscr{P}(\Omega)$ respectively, we have for all $\varepsilon > 0$*

$$\sup_{Q\in B_{\varepsilon,\mathcal{F}}(P)} \operatorname{GAN}_{f,\mathcal{H}}(\mu;Q) \leq \operatorname{GAN}_{f,\mathcal{H}}(\mu;P) + \varepsilon\sup_{h\in\mathcal{H}}\Theta_{\mathcal{F}}(h)$$

**Proof** We have

$$
\sup_{Q\in B_{\varepsilon,\mathcal{F}}(P)} \mathrm{GAN}_{f,\mathcal{H}}(\mu;Q) = \sup_{Q\in B_{\varepsilon,\mathcal{F}}(P)} \sup_{h\in\mathcal{H}} \left( \int_\Omega h dQ - \int_\Omega f^\star(h)d\mu \right)
$$

$$
\stackrel{(1)}{=} \sup_{h\in\mathcal{H}} \sup_{Q\in B_{\varepsilon,\mathcal{F}}(P)} \left( \int_\Omega h dQ - \int_\Omega f^\star(h)d\mu \right)
$$

$$
= \sup_{h\in\mathcal{H}} \left( \sup_{Q\in B_{\varepsilon,\mathcal{F}}(P)} \int_\Omega h dQ - \int_\Omega f^\star(h)d\mu \right)
$$

$$
\stackrel{(2)}{=} \sup_{h\in\mathcal{H}} \left( \int_\Omega h dP + \Lambda_{\mathcal{F},\varepsilon}(h) - \int_\Omega f^\star(h)d\mu \right)
$$

$$
\stackrel{(3)}{\leq} \sup_{h\in\mathcal{H}} \left( \int_\Omega h dP + \varepsilon\Theta_{\mathcal{F}}(h) - \int_\Omega f^\star(h)d\mu \right)
$$

$$
\stackrel{(4)}{\leq} \sup_{h\in\mathcal{H}} \left( \int_\Omega h dP - \int_\Omega f^\star(h)d\mu \right) + \varepsilon \sup_{h\in\mathcal{H}} \Theta_{\mathcal{F}}(h)
$$

$$
= \mathrm{GAN}_{f,\mathcal{H}}(\mu;P) + \varepsilon \sup_{h\in\mathcal{H}} \Theta_{\mathcal{F}}(h),
$$

where (1) holds since we can exchange supremums, (2) is due to Theorem 1, (3) holds since $\Lambda_{\mathcal{F},\varepsilon} \leq \varepsilon\Theta_{\mathcal{F}}(h)$ and finally (4) holds since we can upper bound by taking out supremums. ∎

**Lemma 9** *For any $\mu \in \mathscr{P}(\Omega)$, $h \in \mathscr{F}(\Omega, \mathbb{R})$ we have*

$$\inf_{b \in \mathbb{R}} \sqrt{\mathbb{E}_{\mu(X)}[(h(X) - b)^2]} = \sqrt{\mathrm{Var}_\mu(h)}$$

**Proof** Let $\varphi(b) = \mathbb{E}_{\mu(X)}[(h(X) - b)^2]$ and $S(b) = \sqrt{\varphi(b)}$ and using simple calculus we have

$$S'(b) = \frac{\varphi'(b)}{2\sqrt{\varphi(b)}},$$

and noting that $\varphi(b) > 0$, we can find the minima by solving $\varphi'(b) = 0$ by first noting that

$$\varphi(b) = \mathbb{E}_{\mu(X)}[h^2(X)] - 2b\mathbb{E}_{\mu(X)}[h(X)] + b^2,$$

and so we have

$$\varphi'(b) = 0 \iff -2 \cdot \mathbb{E}_{\mu(X)}[h(X)] + 2b = 0$$
$$\iff b = \mathbb{E}_{\mu(X)}[h(X)].$$

Putting this together yields

$$\begin{aligned}
\inf_{b \in \mathbb{R}} \sqrt{\mathbb{E}_{\mu(X)}[(h(X) - b)^2]} &= \inf_{b \in \mathbb{R}} S(b) \\
&= S\left(\mathbb{E}_{\mu(X)}[h(X)]\right) \\
&= \mathbb{E}_{\mu(X)}\left[\left(h(X) - \mathbb{E}_{\mu(X)}[h(X)]\right)^2\right] \\
&= \sqrt{\mathrm{Var}_\mu(h)}
\end{aligned}$$

∎