[Reviews · NeurIPS 2020]

Review 1

Summary and Contributions: This theoretical paper generalizes DRO to consider any integral probability metric (IPM), providing new insights regarding DRO and regularization. In addition, the paper connects DRO to generative modeling through the use of GANs.

Strengths: The work presents theoretical results that provide a foundation for lots of empirical work that has been developed in the past few years regarding DRO. The connections to GANs are interesting and provide nice fodder for further intersection between the fields.

Weaknesses: Although I find the generalization of DRO to IPMs mathematically satisfying, I would also like some more concrete directions regarding the usefulness of these results to practitioners and the broader community. This can come in the form of discussion at the least but probably also in the form of experiments and empirical evidence if possible. Furthermore, it is unclear how relevant the robustness perspective of GANs is to practitioners. The authors suggest that these results pave the way to certificates of robustness. Perhaps this paper could be made stronger by making some headway in that direction rather than leaving it to future work.

Correctness: I have taken a look at the proofs and nothing egregious jumped out as problematic. The results are all reasonable in the sense that they extend/generalize existing results in the literature (e.g. https://arxiv.org/abs/1604.01446).

Clarity: The paper is well organized and generally straightforward to follow. The commentary around the results is useful and informative.

Relation to Prior Work: The paper grounds itself as a theoretical viewpoint on existing DRO approaches and provides a lens to look at many instantiations through a common framework. The connections to GANs is fairly novel (although as I mentioned in the weaknesses section, it is unclear how the robustness level of a GAN can be used downstream in applications).

Reproducibility: Yes

Additional Feedback: Overall I think this paper has good content that is important to be added to the literature. I am left with a somewhat bitter taste in my mouth that I would have liked to see a little more development regarding the usefulness of the results downstream, either to DRO applications or to GANs. Are there any takeaways on how we can better train discriminators? If so, it would be great for the authors to provide even some small-scale or toy experimental evidence that the connections laid out in this work can be used. I will currently rate this as marginally below the acceptance threshold but am eager and willing to be swayed by the authors to modify my decision based on further discussion and evidence of the usefulness of these results. A minor bug that may need fixing: references 17 and 18 are the same.


Review 2

Summary and Contributions: The paper studies distributionally robust optimization (DRO) with uncertainty sets defined by integral probability metrics (IPMs). The authors derive an exact reformulation of DRO with IPM uncertainty sets (e.g. Wasserstein-1 or MMD) in terms of regularized ERM. The regularizer is similar to but more general than and, due to the exactness of the reformulation, tighter than others in the literature. Later, they also connect IPM DRO with regularization in GANs.

Strengths: I really appreciate the unified view that is possible by treating IPMs generally, and think this is the single strongest aspect of the paper. The unified view will inspire future work connecting the plethora of DRO uncertainty sets studied in the community, which are mostly studied separately. The generalized IPM results are novel to my knowledge, though the individual bounds for each example IPM reflect existing literature. DRO, particularly this form of DRO (uncertainty balls around the empirical distribution) are highly relevant to NeurIPS. Overall, I lean slightly towards reject due to other weaknesses of the paper, but because of the strength of the core unifying results, I would not mind too much if it were accepted.

Weaknesses: The greatest weakness of the paper is probably the GAN section. The motivation for this section seems quite imprecise, and it makes it very difficult for me to judge the section as being significant. I discuss this more in the clarity section. ---- The DRO background context is a bit weak (discussed in prior work section). ---- While I appreciate the generality of the main results, I am a bit concerned that the level of generality is not entirely necessary and may complicate the presentation, e.g., the authors present work for general IPMs, but, as the authors note in lines 225-226, the only concrete IPMs they study are of the simpler form {h : \zeta(h) <= 1}. The authors might consider either discussing other IPMs not of this form, or perhaps presenting simpler results tailored to this type of IPM in the main text and deferring the more general statement to the appendix. It would also help to better illustrate how the new bounds differ from the old, slightly looser bounds. The authors do discuss such examples, starting at line 205, but it would help to have a plot or an experiment or something a bit more concrete.

Correctness: The theory seems appropriately rigorous; I do not doubt its correctness. One note: line 213 caught my attention since for many of the common RKHSes H used, constant functions (i.e. b) will not belong to H. But I suppose this is fine If the domain \Omega is compact.

Clarity: The paper was overall straightforward and easy to understand through section 3. However I found Section 4 a bit unclear: It would be helpful for the authors to front-load discussion on why we want to study the robustness of GANs in this particular way. In the current version of this section of the text, the first mentions of robustness (around line 297) are not at all specific as to why we are studying robustness: "the existing literature is silent on the story of robustness", "Consider now the perspective of distributional robustness." What do we want to be robust to, and why? The closing paragraph of the section is a bit better but is still fairly vague, e.g. what is the meaning of "[..] allowing us to view these methods from a robustness perspective in light of [our results]." It is not until lines 315-321 that the authors get more specific about possible insights from this perspective, e.g. the tradeoffs involved in changing the discriminator set size. The paper could be greatly improved by hinting at this earlier, and, more generally, making statements in this section more precise.

Relation to Prior Work: The authors write at the outset (i.e. line 31) that "DRO studies the objective sup_{Q in U} E_Q[l] where U={Q : d(Q,P) < \ep}" -- this is a great oversimplification, as a significant amount of the DRO literature deals with different kinds of uncertainty sets, e.g. uncertainty sets defined by moment constraints, or uncertainty sets defined to capture causal invariants. The authors should amend that sentence, and, at the very least, cite a few survey papers to point the reader to the broader scope. (for example, Bertsimas, Gupta, Kallus, "Data-Driven Robust Optimization" or Goh, Sim, "Distributionally Robust Optimization and Its Tractable Approximations") The DRO perspective on GAN training and penalties has been studied before in Gao, Chen, Kleywegt, "Wasserstein distributional robustness and regularization in statistical learning" -- the authors should elaborate on how their work differs from this.

Reproducibility: Yes

Additional Feedback: There is a bit of mathematics that goes undefined or is perhaps more abstract than necessary: - in line 103, does it really matter that X is an additive monoid? - in line 239, the authors use the concept of minorization but do not define it.


Review 3

Summary and Contributions: The authors have developed general framework to understand distributionally robust optimization (regularization). The authors first suggest general form of functional penalty called 'F-penalty', and show that the constraint on P lying in IPM-ball can be replaced to penalty-based regularization with F-penalty. The authors investigate the connection between F-penalty and existing penalties with much tight upper-bound. In final section, proposed penalty is applied to f-GAN which gives insight dealing with robustness in GAN objective modeling.

Strengths: Even if i can not deeply understand meaning of the theorem 1, to best of my knowledge, the way the authors to transform left-hand side of theorem 1 into left-hand side is novel. The proof is based on convex analysis (similar to most of recent works), and it seems correct for me. In the middle of section 3, the authors provide the explicit comparisons of \theta_F with existing methods in L:205~223 which clearly shows the improved tightness. In the theorem 3, the proposed penalty successfully plug into conventional f-GANs objective. The DRO of f-GAN objective is bounded by (\theta_F \approx discriminator complexity + f-GANs objective) where various type of \theta_F is shown in table 1.

Weaknesses: I understand this is a theoretical work, and the results and contributions are clear and sound, it is a little frustrating for me because the paper includes no experimental study even with small dataset, no suggestion of tractable algorithms for building robust GANs. While the most recently developed GANs have focused not only on theoretical superiority, but also its quantitative results (better quality of images), i hope to see the empirical connection of the DR model with proposed method to f-GANs shown in theorem 3.

Correctness: The results and contributions are clear and sound for me.

Clarity: Overall, I found this paper to be a nice read.

Relation to Prior Work: The authors clearly discussed contribution of proposed penalty in terms of both difference and improvement.

Reproducibility: No

Additional Feedback:


Review 4

Summary and Contributions: This authors show how distributional robustness with respect to a family of common probability metrics (IPMs) can be rephrased as regularization. They draw connections between their results and existing bounds for the Wasserstein distance among others, and present what their claims mean for robust GANs.

Strengths: * Links to related claims are frequent and clear * Unifies claims across metrics in the family of IPMs * Upper bound on robust GAN objective, split into the standard GAN objective + complexity of the discriminator set

Weaknesses: The paper is well-written, and no drastic weaknesses appear outside of a missing connection between the theoretical claims and machine learning in practice. This link is not crucial, but I thought it would be included.

Correctness: No errors show up to me.

Clarity: Yes, despite the complexity of the topic the sentences are accessible.

Relation to Prior Work: It is clear.

Reproducibility: Yes

Additional Feedback: As someone whose experience lies further away from this work, I thought it might be useful to point out questions that remain after a couple reads of the paper. The claims seem to be of interest to a broader audience and my commentary is intended towards that goal. Below is are two sets of claims from the paper. The first are ones which I had questions about and the second include the area I believed supported the claim most strongly. If this differs from what you intended, it may be useful to rewrite support for that claim. "permitting us to provide untried robustness perspectives for existing regularization schemes" - What are the "untried robustness perspectives"? Is it the relationship between robustness and regularization? Or the ability to interpret a broader set of probability metrics in terms of the regularization they afford? "allow us to give positive results and robustness perspectives" - The phrase "positive results" was confusing to me - is this positive in the sense that the theory supports the validity of these approaches? "Robustness perspectives" also seems vague here. "While these are important to understand, they, however, do not give immediate consequences for machine learning" - This sentence tells me that there are immediate consequences to machine learning from this work. The impacts I picked up on are the importance of discriminator complexity in upper-bounding the robust GAN objective and the potential for robustness certification for GANS. The paper may benefit from a further motivation of robustness certification. - I am curious about how these results affect GAN development in practice. I think it'd be good to clarify if this paper (1) unifies many claims and provides theoretical support for popular practices in GAN development, existing directions of research or if the or (2) offers a theoretical backing for one set of practices over another or (3) something else. "We find that these new penalties are related to existing penalties in regularized critic losses" - This is supported by Theorem 2 "we present a necessary and sufficient condition such that 64 ΛF coincides with ΘF , yielding equality" - This is supported in Lemma 2 "reveals an intimate connection between distributional robustness and regularized binary classification" - Lines 251-253 "the first analysis of robustness for f-GANs with respect to divergence-based uncertainty-sets" - Can't comment - not familiar with the work on robustness of GANs

[Author Response · NeurIPS 2020]

We would like to thank all the reviewers for their efforts in reading the manuscript and providing feedback. The reviewers have appreciated our central objectives such as bridging DRO to regularized machine learning in a "unified view" (R2), providing "foundation for lots of empirical work" (R1), and to "inspire future work connecting the plethora of DRO uncertainty sets" (R2) which are "highly relevant to NeurIPS" (R2). We particularly thank the reviewers for making us aware that we failed in explaining two aspects of our work with sufficient clarity, which we are happy to include in the updated version. We state these here and will reference them in our individual reviewer rebuttals below. We will be using citation references from the main submission.

**1 Concrete directions for practitioners:** Although this is deferred to future work, an immediate consequence in this direction would be robust certification, based on the black-box verification framework in [14], which is briefly mentioned at the end of Section 3. We outline how Corollary 1 directly implies a certificate for practitioners: Given a binary classifier and reference distribution $\rho$, one can compute $\mathbb{E}_{\rho(X)}[h(X)] - \epsilon \Theta_{\mathcal{F}}(-h)$ and check if this value is $\geq 0$. Using Definition 2.2 of [14] and Corollary 1 of our work, if this value is $\geq 0$ then this certifies that the classifier is robust to $\mathcal{F}$-IPM perturbations around $\rho$. This follows from the fact that Corollary 1 (using $-h$) implies $\mathbb{E}_{\rho(X)}[h(X)] - \epsilon \Theta_{\mathcal{F}}(-h) \leq \inf_{Q \in B_{\epsilon, \mathcal{F}}(\rho)} \mathbb{E}_{Q(X)}[h(X)]$ and positivity of the term on the right is precisely the condition laid out in Definition 2.2 of [14]. We will include this immediate consequence in the updated manuscript.

**2 Relevance of the GAN robustness results:** The main takeaway from the results presented in Section 4 is to advocate the use of regularized discriminators when training GANs. In particular, we show that the generative distribution learned using regularized discriminators gives guarantees on the worst-case perturbed distribution (robustness). This is particularly relevant for the robustness community since lines of work [55, 8, 60, 59, 28, 26, 41, 47, 48, 24, 57, 42] implement GANs as a robustifying mechanism by training a binary classifier on the learned GAN distribution. In light of our results, learning a binary classifier using a GAN (trained with regularized discriminators) as a downstream task implies this classifier will consequently be robust. For the GAN community, our finding complements existing empirical evidence that shows benefits of regularized discriminators such as the Wasserstein-, MMD-, and Sobelov-GAN and other discriminator regularizers outlined in [19]. Furthermore, another subtle benefit of Theorem 3 is it shows how a DRO result can be applied to the GAN objective. This paves the way for future developments of DRO to be applied to GANs and consequently helps bridge these two communities.

**Reviewer 1:** Thank you for your feedback and pointing out how our work provides foundation for many empirical work and the importance to be added to the literature. Regarding your points on the usefulness, we have made some headway on how a robust certificate is immediate in **1** and outline the relevance of Section 4 to both the GAN and robustness community in **2**. We really appreciated your feedback, which will be included in the updated version, and are confident that it will improve the paper by virtue of the changes outlined in **1+2**.

**Reviewer 2:** Thank you for your feedback and support of the unified view we present. Regarding the significance of the GAN section, we have outlined this in **2**. More specifically, the motivation for studying GAN robustness comes from lines of work that use the distribution learned by a GAN to train a classifier or to attack (cited above in **2**). In this context, our results allow us to understand how robust a GAN is and what makes them more robust. In particular, our results link this to regularizing the discriminator - validating methods that use GANs for these purposes. We appreciate your comment on pointing out the motivation and to hint on this earlier, which we believe will strengthen our paper. Thank you for the definitions we missed and related work regarding the DRO results, we will amend the statement and include these references. We will also include the paper you mentioned, which focuses on Wasserstein distances and supplements the use of restricted discriminators in Wasserstein GANs (WGAN). In contrast, we develop results for the IPMs (including Wasserstein) and supplement a large family of existing GANs that use restricted discriminator sets (including WGAN). Indeed, since well-known IPMs are of the form $\{h : \zeta(h) \leq 1\}$ (such as Wasserstein distance, Total Variation, MMD, Dudley, etc.), we will take your advice of delegating the general statement to the Appendix as this is only a slight change in notation yet retains the generality of the story and improves presentation - thank you.

**Reviewer 3:** Thank you for your feedback and support of the paper, including comments regarding the novelty and improved tightness of the results we present. Indeed, in terms of empirical connection, the main insight of our results is to supplement existing empirical work that use regularized discriminators in GANs (for example Wasserstein-, MMD- and Sobelov-GANs) , and contributing more largely to this narrative of robustness through regularization. A more direct practical ramification is outlined in **1** above and **2** for more detail regarding the takeaway from our GAN result, such as the robustness guarantees of a classifier trained using GANs.

**Reviewer 4:** Thank you for your feedback and support of the paper. The 'untried robustness perspectives' refers to linking regularization towards robustness. Indeed, the 'positive results' refer to the validity of the approaches. Regarding clarity of our GAN results, it is to provide theoretical support for many popular practices in GAN development as you mention in (1), which we will clarify in the updated version as outlined in **2**.

[Meta-Review · NeurIPS 2020]

This work provides general results in distributionally-robust optimization for any integral probability metric, as well as some discussion of the consequences of these results in f-GANs. The former contribution seems to be a solid improvement to the literature, while the latter is more tentative but might both be relevant to some uses, as well as providing a perspective on the importance of regularization in GAN settings. The paper as submitted had significant clarity issues in terms of the consequences of the work (particularly the GAN section, but the DRO section as well). The author response did a lot to clarify this, in particular bringing up the connection to adversarial robustness. (Although general, it does sometimes reduce to not-too-surprising outcomes; e.g., in the Wasserstein case, it reduces to simply bounding standard adversarial robustness via the Lipschitz constant.) I think, then, that the paper does meet the bar for inclusion at this NeurIPS. I strongly recommend, however, that you make changes corresponding to the discussion in the rebuttal to the final version of the paper, which should make the paper much more accessible and hence useful to the community. [In response to your email, which was sent when I no longer had the ability to email you back: indeed, the reviewers had not updated their reviews at the time the updated reviews were released. I did, though, carefully read your (thorough) author response and take it into account in decision-making.]